# Tiered sympathetic control of cardiac function revealed by viral tracing and single cell transcriptome profiling

Sachin Sharma[1,2†], Russell Littman[3,4,5†], John D Tompkins[1,2], Douglas Arneson[6], Jaime Contreras[1,2], Al-Hassan Dajani[1,2], Kaitlyn Ang[1,2], Amit Tsanhani[1], Xin Sun[7], Patrick Y Jay[8], Herbert Herzog[9], Xia Yang[3,4,5], Olujimi A Ajijola[1,2]*

[1]UCLA Neurocardiology Research Center of Excellence, Los Angeles, United States; [2]UCLA Cardiac Arrhythmia Center, Los Angeles, United States; [3]UCLA Bioinformatics Interdepartmental Program, Los Angeles, United States; [4]UCLA Integrative Biology and Physiology, Los Angeles, United States; [5]UCLA Quantitative and Computational Biosciences, Los Angeles, United States; [6]UCSF Bakar Computational Health Sciences Institute, San Francisco, United States; [7]University of California, San Diego, San Diego, United States; [8]Alnylam Pharmaceuticals, Cambridge, United States; [9]Neuroscience Division, Garvan Institute of Medical Research, St. Vincent's Hospital, Darlinghurst, Australia

**Abstract** The cell bodies of postganglionic sympathetic neurons innervating the heart primarily reside in the stellate ganglion (SG), alongside neurons innervating other organs and tissues. Whether cardiac-innervating stellate ganglionic neurons (SGNs) exhibit diversity and distinction from those innervating other tissues is not known. To identify and resolve the transcriptomic profiles of SGNs innervating the heart, we leveraged retrograde tracing techniques using adeno-associated virus (AAV) expressing fluorescent proteins (GFP or Td-tomato) with single cell RNA sequencing. We investigated electrophysiologic, morphologic, and physiologic roles for subsets of cardiac-specific neurons and found that three of five adrenergic SGN subtypes innervate the heart. These three subtypes stratify into two subpopulations; high (NA1a) and low (NA1b and NA1c) neuropeptide-Y (NPY) -expressing cells, exhibit distinct morphological, neurochemical, and electrophysiologic characteristics. In physiologic studies in transgenic mouse models modulating NPY signaling, we identified differential control of cardiac responses by these two subpopulations to high and low stress states. These findings provide novel insights into the unique properties of neurons responsible for cardiac sympathetic regulation, with implications for novel strategies to target specific neuronal subtypes for sympathetic blockade in cardiac disease.

*For correspondence: OAjijola@mednet.ucla.edu

†These authors contributed equally to this work

## Editor's evaluation

This is a key paper reporting the identification of sympathetic ganglion subtypes with differing neuropeptide Y levels, gene expression, electrophysiological properties, and roles in cardiac sympathetic excitation of the heart. Molecular and cellular level findings are validated by their physiological effects. This is a research programme of fundamental and potentially therapeutic importance.

## Introduction

Cardiac autonomic regulation is exerted by reflex loops consisting of afferent vagal and spinal afferent inputs to the nervous system, and descending control by efferent parasympathetic and sympathetic

control (**Ardell et al., 2016**; **Shivkumar et al., 2016**). Postganglionic sympathetic neurons responsible for direct cardiac sympathetic neurotransmission are located within the cervico-thoracic sympathetic chain, particularly the stellate ganglia and middle cervical ganglia (**Hanna et al., 2017**; **Rajendran et al., 2019**). These stellate ganglion neurons (SGNs) release neurotransmitters that regulate all aspects of cardiac function including inotropy and chronotropy. However, following cardiac injury such as myocardial infarction (MI) and heart failure (HF), SGNs undergo profound structural, neurochemical, and electrophysiologic remodeling, contributing directly to progressive cardiac dysfunction and lethal ventricular arrhythmias such as ventricular tachycardia/fibrillation (**Ajijola, 2015**; **Ajijola et al., 2017**; **Alston et al., 2011**; **Gardner et al., 2016**; **Han et al., 2012**). As a result, anti-adrenergic therapies such as beta-adrenergic receptor and neurohormonal blockers remain cornerstone treatments for MI and ventricular arrhythmias (VT/VF) (**Gardner et al., 2016**; **Sana, 2018**). Further, interventional and surgical therapies targeting the stellate ganglion (e.g. percutaneous anesthetic blockade and surgical sympathetic denervation, respectively) are increasingly applied clinically (**Ajijola et al., 2012**; **Bourke et al., 2010**; **Meng et al., 2017**).

Despite strong clinical data supporting therapies targeting SGNs to treat cardiac disease, the efficacy of available therapies remains limited, particularly by systemic side effects such as hypotension or renal dysfunction (**Lisi et al., 2020**; **Mets et al., 1992**; **Rich, 2012**; **Sica and Black, 2002**; **Tang and Maroo, 2007**). Little is known about the subtypes and properties of neurons that innervate the heart, compared to neurons innervating other tissue beds, for example, skin and paw, that might facilitate the development of more specific targets for therapeutic purposes (**Li and Dahlström, 2008**; **Taniguchi et al., 1994**).

Prior work investigating SGN subpopulations focusing on nipple and piloerector neurons used single cell RNA sequencing (scRNAseq) to identify eight subpopulations of neurons within the stellate ganglion (**Furlan et al., 2016**) in mice. Specifically, five noradrenergic, two cholinergic, and one gluta-minergic neuronal subtype were identified, each with unique combinations of transcription factors, neuropeptides, and ion channels. It remains unknown whether a specific subtype of neurons innervates the heart, or whether various subtypes contribute to cardiac adrenergic control.

This study used retrograde viral tracing by injecting adeno-associated viruses (AAVs) into the myocardium of mice to identify and label cardiac-specific SGNs with the reporter green fluorescent protein (GFP). In contrast, the front paws were injected with AAVs encoding the tdTomato fluorescent protein. Ganglia were collected after 4 weeks and subjected to dissociation and scRNAseq, with cells containing GFP or tdTomato able to distinguish cardiac vs. paw-innervating neurons. We found that the heart vs. paws are innervated by three vs. four subtypes of noradrenergic neurons respectively, largely segregated by neuropeptide Y (NPY expression). These subtypes were also identified in porcine and human stellate ganglia. Clinical data implicate NPY signaling in MI, progression of left ventricular dysfunction, ventricular arrhythmias and HF, thus we focused on characterizing neurons based on NPY expression (**Ajijola, 2015**; **Ajijola et al., 2020**; **Kalla et al., 2020**). We find that NPY +vs. NPY- neurons exhibit unique transcriptional profiles, along with unique morphological, neurochemical, and electrophysiologic properties. Functional studies in NPY-expressing neurons ablated mice indicate that NPY is required for maximal cardiac sympatho-excitation. Collectively, these findings shed light on cell-specific cardiac adrenergic regulation with implications for novel therapeutic targeting of adrenergic signaling following cardiac injury.

## Results

### Three unique stellate ganglion neuronal subtypes innervate the heart

We performed scRNAseq on stellate ganglion neurons from eight retrograde-labeled C57BL/6 mice. We identified cardiac innervating neuronal subpopulations in stellate ganglia after labeling mice retrogradely from the heart using AAV-GFP and from front paws using AAV-tdTomato as a control (**Figure 1A**, left panel, **Figure 1—figure supplement 1A**, left panel), *respectively*. We observed regional specificity in stellate ganglia for cardiac vs. paw innervating neurons. Most cardiac-labeled neurons were localized in the cranio-medial region (cardiac pole) of the stellate ganglion while paw labeled neurons were distributed across the stellate ganglion (**Figure 1A**, right panel, **Figure 1—figure supplement 1A**, right panel). We validated the retrograde labeling of SGNs from the heart by staining AAV-GFP injected heart-sections with antibodies against the pan-neuronal marker PGP9.5.

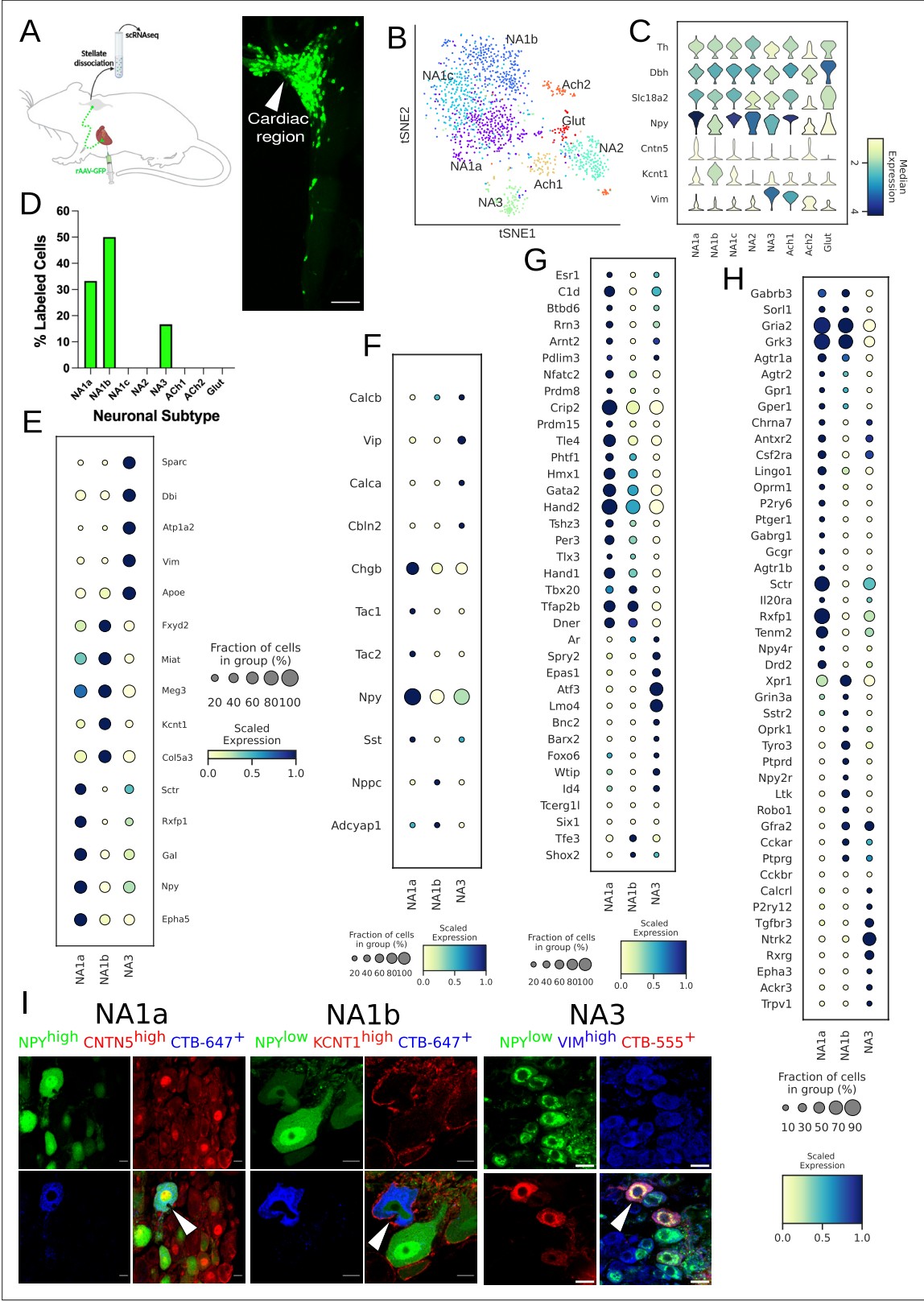

**Figure 1.** Distribution of cardiac-specific neuronal sub-population in the mouse stellate ganglia. (**A**) Experimental overview, diagrammatic representation of tracing from cardiac (AAV-GFP) neurons to the stellate ganglion, followed by single-cell RNA sequencing (scRNAseq) analysis (left). The CTB labeled right stellate ganglion shows the regional specificity of cardiac neurons (right) in the stellate ganglion. (**B**) tSNE plot visualizes eight Louvain identified neuronal subpopulations with prior annotations. (**C**) Violin plot of key marker genes used to validate the sympathetic cardiac

*Figure 1 continued on next page*

*Figure 1 continued*

neuronal subtypes. (**D**) Distribution of cardiac neuronal subpopulations in stellate ganglia. n=8 mice. (**E**) Dot-plot shows the transcriptomic profile in each cardiac subtype. Dot-plot shows relative expression of neuropeptides (**F**); transcription factors (**G**); and receptors (**H**) associated with subtypes. (**I**), Immunohistochemistry validation of cardiac innervating neuronal subtypes (NA1a, NA1b, NA3). *Scale bar: 10 and 20* µm.

The online version of this article includes the following figure supplement(s) for figure 1:

**Figure supplement 1.** Paw innervating neuronal subpopulation in the mouse stellate ganglion.

**Figure supplement 2.** Validation of retrograde tracing of AAVs from myocardium (Heart injections).

**Figure supplement 3.** scRNAseq quality control.

**Figure supplement 4.** Neuronal subtypes in the bilateral stellate ganglion tissues.

**Figure supplement 5.** Transcriptomic profiles of each neuronal subtype in stellate ganglia.

**Figure supplement 6.** Immunohistochemistry validation of bilateral stellate ganglion neuronal subtypes identified from scRNAseq data analysis.

**Figure supplement 7.** Accuracy of retrograde labeling of cardiac- and paw-innervating neuronal subtypes in stellate ganglia.

In multiple cardiac cross-sections, we found nerve fibers co-labeled with GFP and PGP9.5 antibodies (*Figure 1A, right panel, Figure 1—figure supplement 2*). Since this expression generally requires retrograde transport of AAV to the soma from nerve endings in the heart, GFP expression, and antegrade transport of GFP to cardiac nerve fibers, it confirmed that we successfully labeled postganglionic neurons in the stellate ganglion by cardiac injections.

We clustered scRNAseq data, identified cell types based on their canonical markers, and separated out the neuronal population based on cluster markers *Dbh, Th,* and *Snap25* (*Figure 1—figure supplements 3 and 4*; *Supplementary file 1*; *Supplementary file 2*). We identified eight subtypes of stellate ganglion neurons: five dopaminergic, two cholinergic, and one glutamatergic neuronal subtype in the stellate ganglia (*Figure 1B–C, Figure 1—figure supplements 4–6*). In our scRNAseq studies, we identified labeled cells in the stellate ganglion either as cardiac- or paw- innervating neuronal subtypes based on the presence of GFP or tdTomato transcript, respectively. We found a total of three GFP-expressing (Cardiac) neuronal subpopulations NA1a, NA1b, and NA3 and four tdTomato-expressing neuronal subtypes NA1a, NA1b, NA1c, and NA3 (*Figure 1D, Figure 1—figure supplement 1B*). Despite the overlap in subtypes, we did not find the same neuron labeled by retrograde tracers from the heart and paw.

Interestingly, we found that similar neuronal subpopulations (NA1a, NA1b, and NA3) innervate both heart and front paws, however the front paws are innervated by an additional neuronal subpopulation NA1c (*Figure 1D, Figure 1—figure supplement 1B*). This suggests that similar subsets of neurons in the stellate ganglion are responsible for neural control of a variety of tissues, rather than tissue-specific subtypes. We observed significant differences in the transcriptome of each neuronal subtype (*Figure 1E, Figure 1—figure supplements 1C and 4D–F*), with distinct relative expression levels of secreted neuropeptides, transcription factors, and receptors (*Figure 1F–H, Figure 1—figure supplement 1E–G*).

Neuropeptide-Y is a potent modulator of cardiac function in health and disease (*Walker et al., 1991*). Our scRNAseq data revealed that cardiac innervating neuronal subtypes can be distinguished based on *Npy* expression (*Figure 1E*). To validate the neuronal subtypes and their localization in heart and paw neurons (*Figure 1I, Figure 1—figure supplement 1H*), we performed retrograde labeling from the heart and paw using CTB conjugates to either Alexa-555 or –647 in wildtype or *Npy*-hrGFP transgenic mice. We then performed immunohistochemistry (IHC) using antibodies against markers specific to each transcriptomic subtype of neurons.

We performed combinatorial IHC as follows; $Npy^{high}$, $Cntn5^{high}$ for NA1a, $Npy^{low}$, $Kcnt1^{high}$ for NA1b, and $Npy^{low}$, $Vim^{high}$ for NA3, and $Npy^{low}$, $Sctr^{high}$ for NA1c (*Figure 1I, Figure 1—figure supplement 1H*). As illustrated in the figure, we identified all expected markers in cardiac and paw neuronal subtypes and also carried out combinatorial immunohistochemistry for the non-cardiac innervating neuronal subtypes using markers for each subtype. We found less than ≤6% of cells co-labeled with the cardiac retrograde label (CTB-647) when stained for these combinations of markers (*Figure 1I, Figure 1—figure supplement 7*). Very few cells identified by IHC as non-cardiac were labeled by retrograde cardiac tracing further supporting the accuracy of our AAV-labeling and scRNAseq approach.

Additionally, we validated the presence of these cardiac-innervating neuronal subpopulations in porcine and human stellate ganglia (*Figure 2A and B*), respectively. We used the following combination

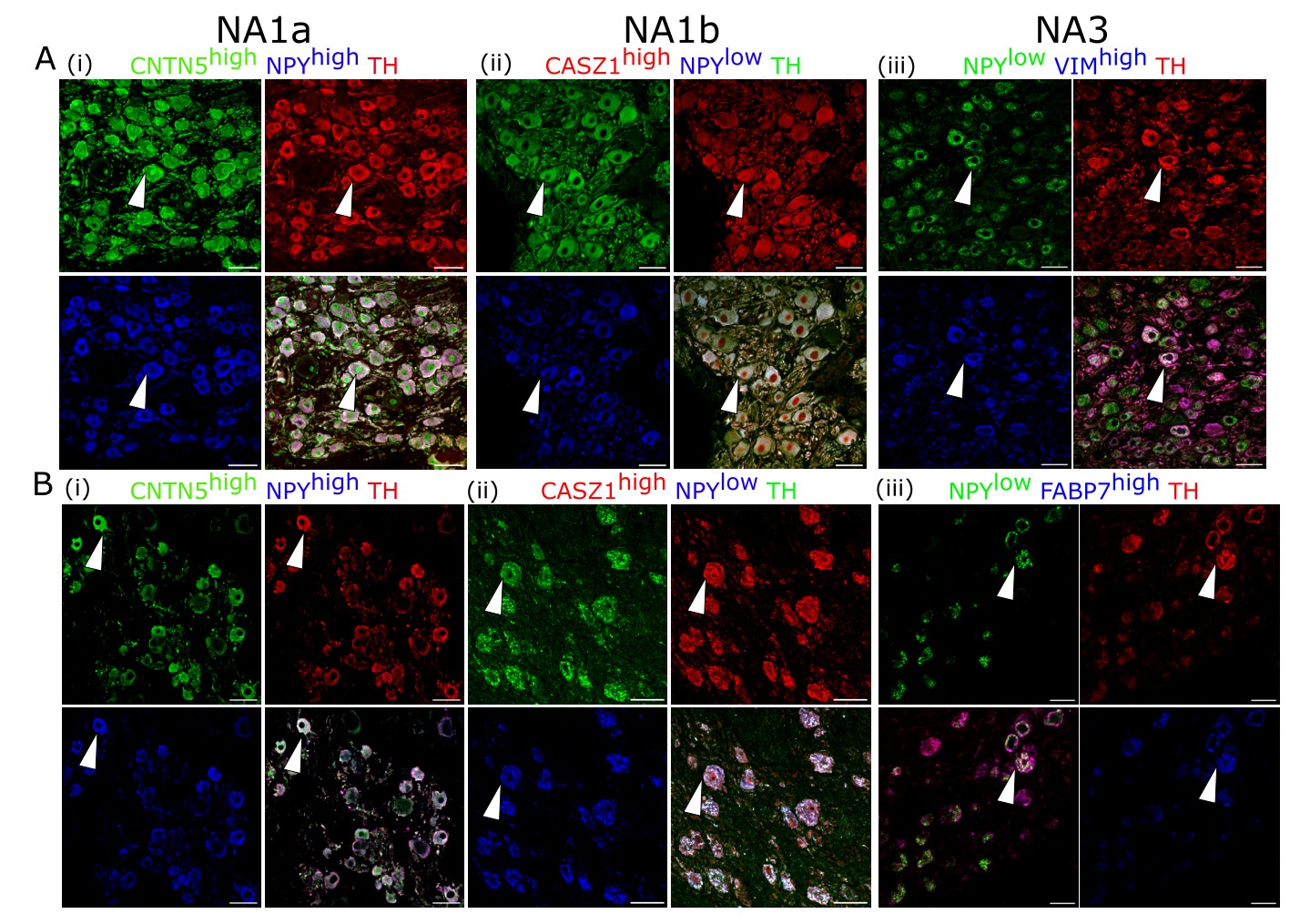

**Figure 2.** Cardiac clusters were also validated in porcine (**A**) and human (**B**) stellate ganglia using immunohistochemistry by combining the specific marker genes for each neuronal subtype. (**A, B**) (**i**) NA1a neurons express high NPY and CNTN5. (**A, B**) (**ii**) NA1b subtype has low NPY and high CASZ1 expression level. (**A**) (**iii**) NA3, similar to NA1b, has low NPY but uniquely higher Vimentin expression in porcine tissues. (**B**) (**iii**) NA3, like NA1b, expresses low NPY and high FABP7 expression in humans. Scale bar: *50* μm. n=3 samples.

of subtype-specific markers; *NPY*high, *CNTN5*high (NA1a), *NPY*low, *CASZ1*high (NA1b), and *NPY*low, *VIM*high (NA3) in porcine stellate ganglia (***Figure 2A***) and *NPY*high, *CNTN5*high (NA1a); *NPY*low, *CASZ1*high (NA1b); and *NPY*low, *FABP7*high (NA3) in human stellate ganglia (***Figure 2B***). Identification of these subtypes in porcine and human stellate ganglia suggests that similar subtypes of neurons may innervate the heart in pigs and humans.

## Cardiac subtypes differentially express NPY

NPY is a co-transmitter released by sympathetic nerve terminals along with norepinephrine to dynamically modulate cardiac function (***Tan et al., 2018***; ***Zhu et al., 2016***). Elevated circulating NPY levels are associated with adverse outcomes in chronic heart-failure patients. Our scRNAseq findings demonstrate high *Npy* expression in NA1a neurons, but lower expression in NA1b and NA3, independent of cardiac (or paw) innervation (***Figure 3A***, ***Figure 3—figure supplement 1A***).

Furthermore, we identified 114 genes positively and 527 genes negatively associated with *Npy* (FDR <0.05; methods) for cardiac subtypes. We show the top 10 positively and negatively correlated genes with *Npy* in cardiac subtypes and in paw subtypes (***Figure 3B***, ***Figure 3—figure supplement 1B***). We confirmed previously known associations between *Npy* and *Gal* (Pearson correlation: 0.64;

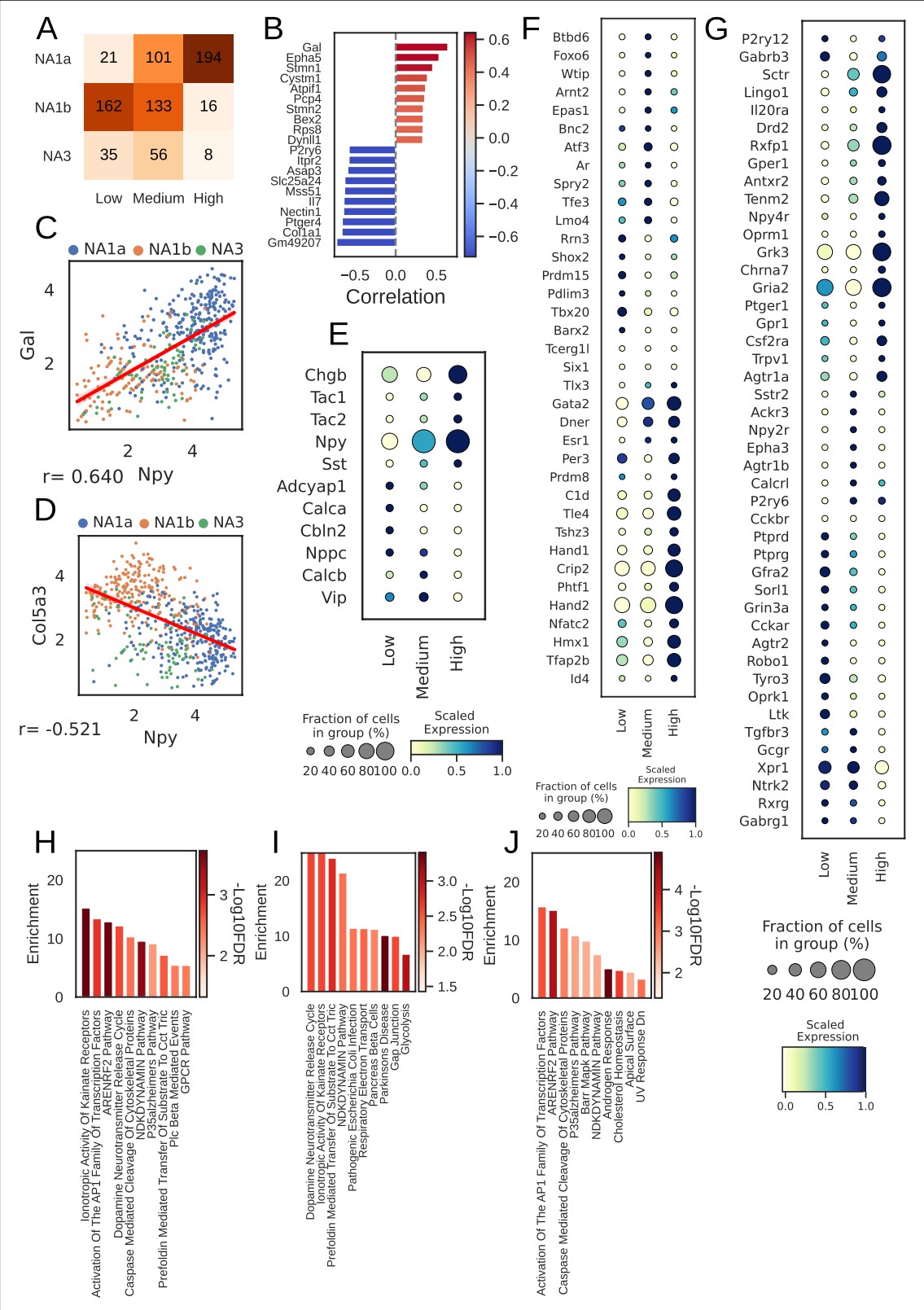

**Figure 3.** Npy gene distribution and the associated genes in cardiac neuronal subtypes. (**A**) Distribution of subtypes across quantiles of *Npy* expression. (**B**) Top 10 positively (red) and negatively (*Lake et al., 2016*) associated genes with *Npy*. Intensity of the color indicates the strength of the Pearson correlation coefficient. (**C, D**) Scatter plots show the positive correlation of Gal (**C**) and, negative correlation of Col5a3 (**D**) with *Npy* expression on the log scale. The dots in the scatter plots are colored to represent each cardiac neuronal subtype. Cells containing zeros for either gene were removed.

*Figure 3 continued on next page*

*Figure 3 continued*

(**E, F, G**) Dot-plots show relative expression of neuropeptide (**E**), receptors (**F**), and transcription factors (**G**) in cells with low, medium, and high Npy expression. (**H**) Top 10 combined; (**I**) Positively and (**J**) Negatively enriched pathways associated with *Npy*. n=8 mice.

The online version of this article includes the following figure supplement(s) for figure 3:

**Figure supplement 1.** *Npy* distribution and associated genes in paw innervating neuronal subtypes.

*Figure 3C*; *Herring et al., 2012*) and uncovered new correlations such as that between *Col5a3* and *Npy* (Pearson correlation: –0.52; *Figure 3D*) in cardiac neuronal subtypes.

Next, we revisited the finding of differential NPY expression in cardiac neuronal subtypes. Interestingly, most neurons in the NA1b or NA3 subtypes were NPY low or NPY negative (*Figure 3A*). This is surprising, given the dogma that NPY and catecholamines exist in the same cells, but NPY-containing granules are released during high stress states, while only catecholamines are released at low stress states (*Cavadas et al., 2002*; *May et al., 1995*).

To further understand how cardiac neurons might differ based on NPY expression, we stratified cardiac subtypes by *Npy* expression level (low/negative, medium, and high) and found unique expression patterns of neuropeptides, transcription factors, receptors, and ion channels corelated with NPY expression level (*Figures 3E–G and 4A–D*, *Figure 3—figure supplement 1C-F*). We corroborated this by performing gene set enrichment analysis on the combined, positive, and negative set of *Npy* associated genes (see Methods).

This analysis revealed that biochemical pathways associated with high *Npy* expression included dopamine neurotransmitter release cycle (catecholamine biosynthesis and release), inotropic activity of kainate receptors (excitatory glutaminergic receptors), and respiratory electron transport (mitochondrial metabolism and function) suggesting that NPY expression is associated with neuronal metabolism and excitability (*Figure 3H–J*, *Figure 3—figure supplement 1G-I*). These findings suggest that the high *Npy* expressing neuronal subtype NA1a and lower expressing subtypes, NA1b and NA3, are indeed distinct cardiac-innervating neuronal subtypes in stellate ganglia.

Next, we sought to immunohistochemically confirm that NPY +and NPY- neurons indeed innervate the heart as suggested by our scRNAseq data. We coupled this question with another, do NPY +versus NPY- neurons innervate disparate regions of the heart? To answer both questions, we performed retrograde labeling from the apex (using CTB-555) and base (using CTB-647) in *Npy*-hrGFP mice (*Figure 4A*), which expresses humanized Renilla Green Fluorescent protein under control of the *Npy* promoter. First, we found that both NPY +and NPY- neurons in stellate ganglia innervate the heart consistent with data from neurons traced and subjected to scRNAseq. Next, we found that neurons traced from the apex and base colocalized to the craniomedial pole of the stellate ganglion (*Figure 4B*) and were admixed, suggesting that there is no spatial representation of cardiac sympathetic neurons in the ganglion.

Next, we examined the subtype distribution of neurons (i.e. *Npy* +vs. *Npy*-, *Figure 4F–H*) innervating the apex (CTB-555+) or base (CTB-647+). We observed that most neurons across both injected sites were *Npy* +compared to *Npy*- (71.8 ± 20% vs 27.7±20% for apex and 66±22% vs 34 ± 22% for base, p<0.001 for both). Further, some neurons were labeled both from apex and base, suggesting these neurons subtend a broad myocardial region. Within this population of neurons, a greater percentage of *Npy* +neurons were labeled compared to *Npy*- (*Figure 4F*, p<0.001). Taken together, these findings confirmed that the heart is innervated by both *Npy* +and *Npy*- neuronal subtypes without a particular spatial representation in the ganglion, although *Npy* +neurons are predominant. Further, these findings demonstrate that *Npy* +and *Npy*- neurons innervate the apex only, base only, and both. Interestingly, when comparing cardiac to paw neurons across animals, ~70% of cardiac-innervating neurons express high *Npy* (*Figure 4I*), compared to 30% innervating the paw.

Next, we assessed whether *Npy* expression was associated with morphologic differences in cardiac-specific neurons identified by retrograde tracing. We found that the soma of Npy + neurons were physically larger than those of *Npy*- neurons (*Figure 4J*). Mean soma size for Npy + vs. *Npy*- neurons was 546±252 µm$^2$ vs. 342±170 µm$^2$ (mean ± SD, p<0.0001). This suggests that compared to *Npy*- neurons, Npy + cardiac neurons may exhibit greater number of synapses on dendrites and axon terminals (*Liu, 2004*; *Pierce and Milner, 2001*), greater speed of action potential propagation and increased metabolic demands (*Brown et al., 2008*; *Laughlin et al., 1998*; *Wang et al., 2008*). We therefore sought to investigate whether Npy + vs Npy- neurons exhibited differential electrophysiologic properties.

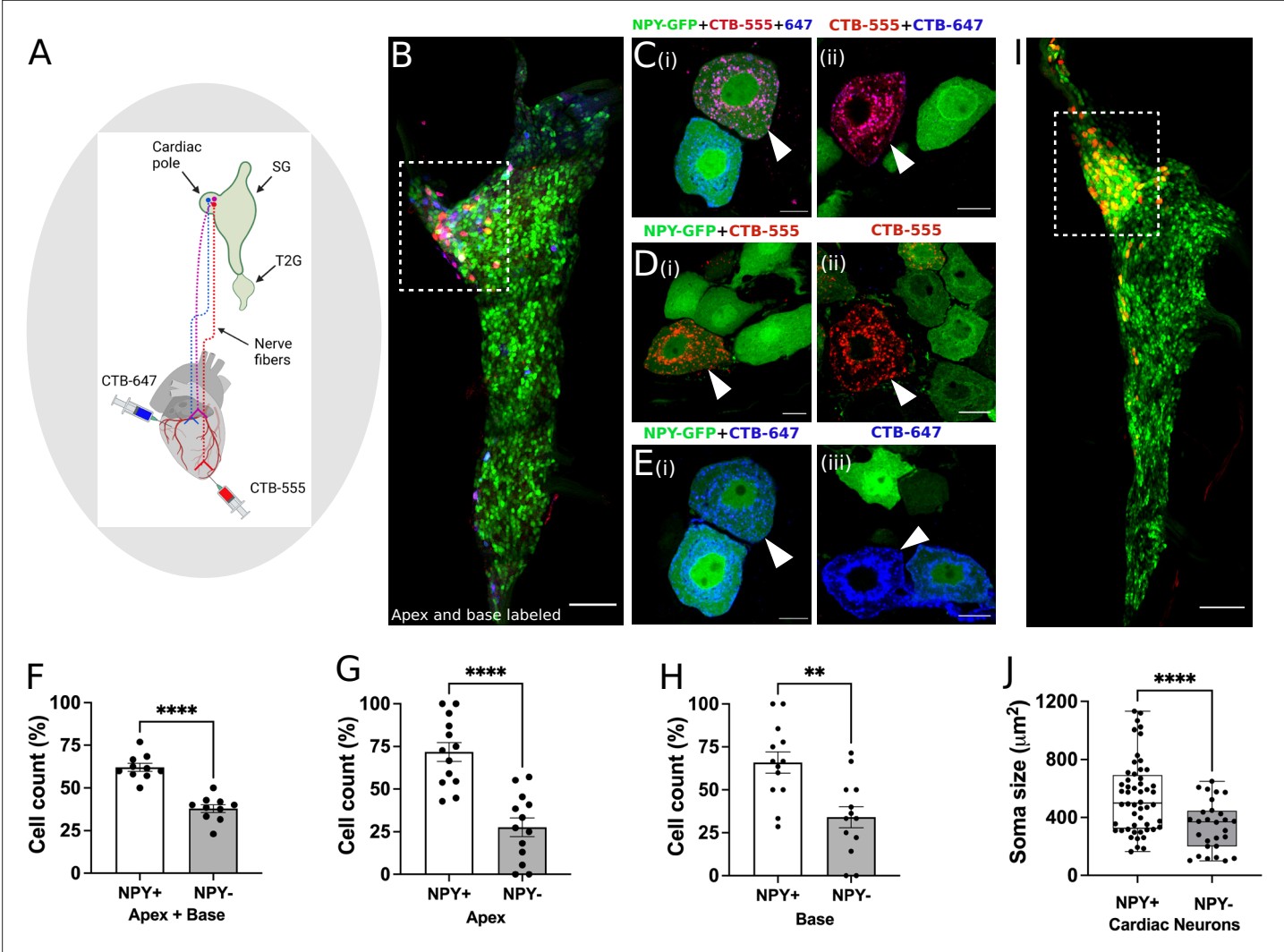

**Figure 4.** Distribution of NPY neurons innervating disparate regions of the heart. Cholera toxin subunit B (CTB) conjugated Alexa Fluor 555 and 647 (CTB-555 or –647) were injected into the apex and base of the heart, respectively. (**A**) A schematic depiction of the approach used to retrograde label stellate ganglion neurons (SGNs). (**B**) An image of the right stellate ganglia from the heart after CTB-555 and CTB-647 retrograde labeling. Scale bars: 200 μm. (**C,D,E**) High-resolution images of several types of co-labeled cells appearing in the stellate ganglia. Scale bars: 10 μm. (**F**) Cell count for NPY +and NPY- neurons that innervate apex and base regions of the heart simultaneously. NPY +and NPY- neurons that innervate either only the apex (**G**) or base (**H**) heart regions. (**I**) Right stellate ganglia from NPY-hrGFP animals labeled retrograde with CTB-555 showing that most cardiac neurons are NPY expressing neurons. High-GFP expression observed in the cardiac pole (box). (**J**) Soma size measurements for cardiac innervating NPY +and NPY- neurons. n=7 mice/group. Data are shown as mean ± SEM and individual data points in (**F**, **G**, and **H**) represent each stellate ganglion. Normal distribution of data was assessed by Shapiro-Wilk and statistical analyses used Welch's parametric t-test for (**F**), (**G**), (**H**), and (**J**). A box and whisker plot representing data points as individual cells visualizes the trend in soma size for cardiac NPY +and NPY- neurons and Welch's t-test was used for finding the statistical significance in panel (**J**). **=p < 0.01, ****=p < 0.001.

## High *Npy*-expressing cardiac neurons exhibit greater excitability

To determine whether NPY-rich cardiac neuronal subtypes exhibit unique electrophysiologic properties, we examined the transcriptome of these subtypes for ion channel expression. We characterized expression of potassium, sodium, calcium, and proton channels in cardiac subtypes (*Figure 5A–D*), and hypothesized based on previous literature (*Davis et al., 2020*) that identified cardiac subtypes may exhibit unique characteristic cellular electrophysiologic properties. To test this, we studied the electrophysiological properties of *Npy*-expressing neurons in the cardiac pole of the stellate ganglion using the reporter mouse line *Npy*-hrGFP. Cells with (GFP+) and without GFP expression (GFP-), were selectively targeted in ganglion whole-mounts using epifluorescence (*Figure 5E and F*). Resting membrane potential values were –51.91±3.14 mV for GFP +and –52.45±2.45 mV for GFP- cells (p=0.899).

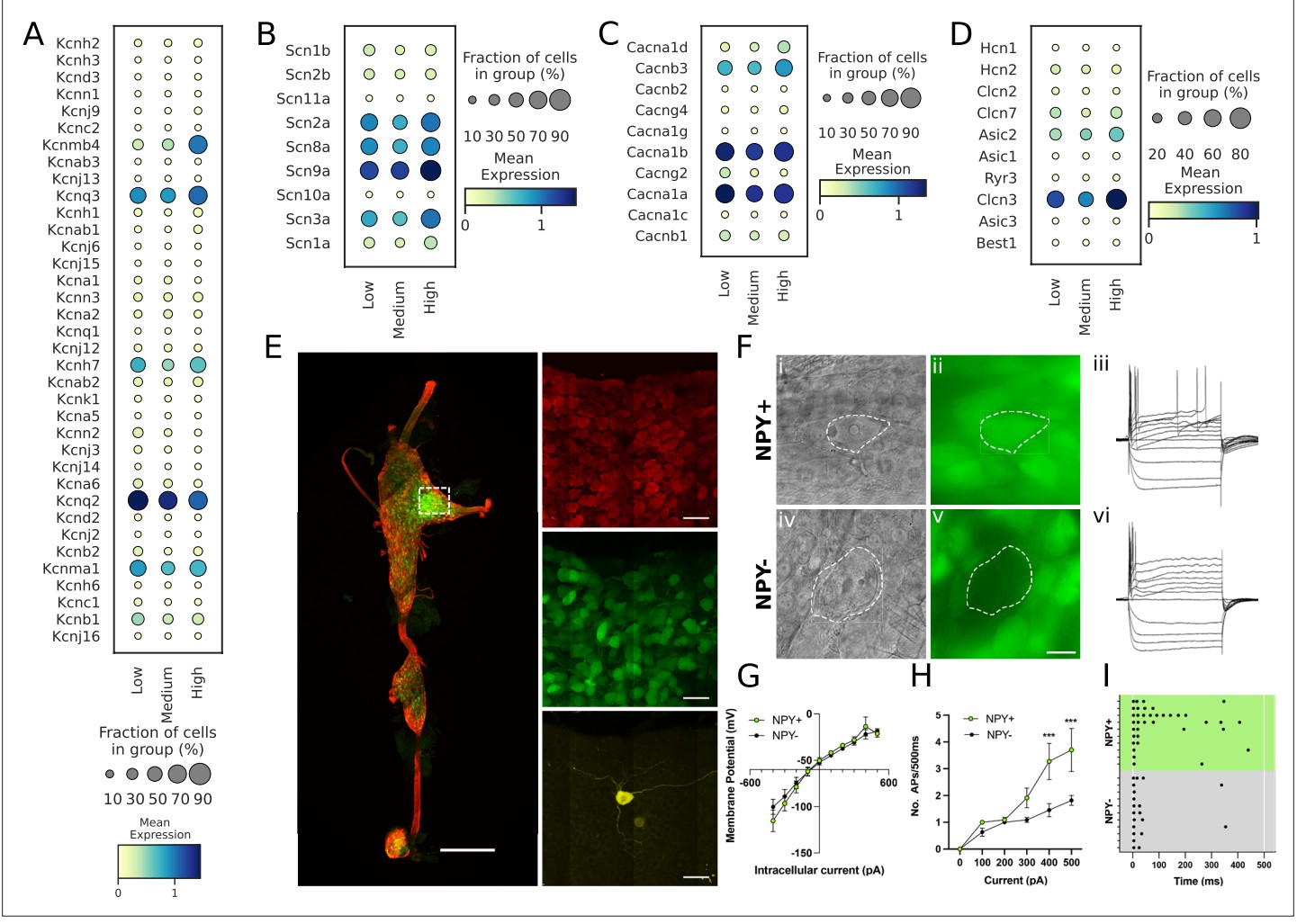

**Figure 5.** Transcriptomic and electrophysiological properties of *Npy*^high vs *Npy*^low/neg neurons. (**A,B,C,D**) Distribution of different ion channels associated with *Npy*^high and *Npy*^low/neg expressing cells: Potassium channels (**A**); Sodium channels (**B**); Calcium channels (**C**); and several other chloride, acid sensing ion channels (**D**) in high, medium, and low *Npy* expressing cells. (**E**) Confocal image of a fixed right stellate ganglion whole mount. PGP9.5 (red), *Npy*-GFP+ (green), and Neurobiotin (yellow). Magnified images from boxed region appear in right hand panels. (**F**) *Npy* + cells identified by epifluorescence during targeting with intracellular microelectrodes. Left panels DIC image (i,iv), right panel GFP (green) (ii),(v). (iii),(vi); Membrane potential tracings from a GFP+ (iii) and a GFP- (vi) cells in response to 500ms hyperpolarizing and depolarizing pulses. Resting membrane potential for both cells was more or less similar to ~−52.45 mV. Dash is 0 mV. (**G,H,I**) Summary electrophysiology data for the *Npy*+ (n=11 mice) and *Npy*- cells (n=11 mice). GFP + cells had similar current-voltage relationship (**G**) but showed greater membrane excitability in response to intracellular depolarization (**H**). Data are shown as means ± SEM; ***p<0.001 compared with *Npy*- cells. The raster plot (**I**) illustrates time of peak action potential amplitude in response to 500ms, 500 pA depolarizing current. Note the greater number of APs and shorter interspike interval in *Npy* + cells that reveal increased neuronal excitability. A two-way ANOVA followed by Sidak's multiple comparison test was used for statistical analyses.

Membrane input resistance values were 106.9±12.63 MOhms in GFP +and 103.5±10.40 MOhms in GFP- neurons (p=0.8391). Current-voltage relationship (***Figure 5G***) were also similar for both cell types over a range of tested currents (–500 pA to +500 pA, Δ100pA) and showed a mostly linear I-V relationship with little rectification ($F_{(9, 128)}$=0.562, p=0.825). In response to depolarizing current steps, both cell types elicited an increasing number of action potentials in response to increasing magnitude of depolarization. While the excitability curves were similar (p=0.5392) up to 300 pA current, Npy + cells showed a significantly (p<0.001) greater number of action potentials than *Npy*- cells in response to 400 pA (NPY+=3.3 ± 0.7 APs, n=11; NPY-=1.5 ± 0.2 APs, n=11) and 500 pA currents (*Npy*+=3.7 ± 0.8 APs, n=10; *Npy*-=1.8 ± 0.2 APs, n=11) (***Figure 5H***). The raster plot (***Figure 5i***) illustrates the shortened interspike intervals of Npy +vs. *Npy*- cells in response to 500ms, 500 pA depolarizing currents. A subset of cells were tested with 700 pA depolarizing steps, which showed greater divergence of

action potential generation in *Npy*+ (12.75±4.3 APs, n=4) vs. *Npy*- (2±0.3 APs, n=5)(p=0.026). Not all cells were tested with depolarizations greater than 500 pA due to the tendency of the high impedance intracellular electrode to occlude with higher current. The peak firing frequency of Npy +and *Npy*- cells is therefore likely underestimated since they were not depolarized to saturation. In response to 500 pA depolarizations, the proportion of neurons firing >3 Aps was significantly (p=0.0046) greater in *Npy*+ (58% tonic) vs *Npy*- (0% tonic). These findings further support the notion that high *Npy*-expressing cardiac sympathetic neurons (NA1a) possess electrophysiological properties distinct from NA1b and NA3 neurons which express low levels or are negative for *Npy*.

## Tiered control of cardiac sympathetic neurotransmission is mediated by *Npy*-expressing neurons in stellate ganglia

Next, we sought to examine how *Npy*-expressing subtypes might impact cardiac function in the physiologic studies. We first examined the role played by NPY + neurons at graded levels of sympatho-excitation (examined by escalating levels of electrical stimulation of the stellate ganglion in mice with or without ablated NPY +neurons). To deplete *Npy* expressing neurons, we injected pAAV-Ef1α-DIO-EGFP+pAAV-flex-taCasp3-TEVp (to express caspase which drives programmed cell death) or control virus (pAAV-Ef1α-DIO-EGFP) in right stellate ganglia of *Npy*-IRES-Cre mice (*Figure 6A*). To confirm ablation of NPY-expressing neurons, we imaged stellate ganglia isolated from DIO-EGFP and DIO-EGFP +Casp3 TEVp injected mice for staining NPY expressing neurons. We observed that NPY expressing neurons were depleted in stellate ganglia from NPY-neuron ablated mice (*Figure 6B and C*). Moreover, in a magnified view, we observed that high NPY expressing neurons localized densely in cardiac pole were highly affected in the Casp3-TEVp injected mice as compared to NPY neurons in other portions of the stellate ganglia (*Figure 6C*). After three weeks of virus incubation in mice, we electrically stimulated right stellate ganglion (RSG) in vivo in mice injected with either DIO-EGFP (n=3) or DIO-EGFP +taCasp3 TEVp (n=5) at 1, 5, 10, and 15 Hz and compared heart rate responses while escalating RSG stimulation frequency (*Figure 6D*). For both groups we escalated stimulation frequency only after the heart rate had returned to baseline following the previous stimulation (*Figure 6E*). Baseline heart rates across the various stimulation frequencies (1, 5, 10, and 15 Hz) remained the same for both groups (*Figure 6E*).

Stimulation frequency and heart rate change showed a linear relationship for both control and NPY neurons ablated mice. However, heart-rate change was significantly lower for NPY neuron ablated mice compared to controls (p=0.0009; *Figure 6F*). We observed maximal significant heart rate change difference for NPY ablated mice (91.061±2.628 bpm) vs. controls (27.017±11.888 bpm) at high stimulation frequencies (10 Hz [p=0.0009] and 15 Hz) but not at low stimulation frequencies (1 Hz and 5 Hz). These findings suggest that *Npy*-expressing neurons are necessary to achieve maximal cardiac sympathoexcitation (i.e. high but not low stimulation frequencies).

To confirm that these findings do not merely reflect depletion of a substantial percentage of stellate ganglion neurons, we examined cardiac responses to escalating stimulation frequencies of right stellate ganglia in *Npy*$^{-/-}$ mice compared to mice expressing *Npy* (i.e. *Npy*$^{+/-}$). In these studies, the *Npy* gene is interrupted but stellate ganglion neurons are otherwise unperturbed. First, we examined *Npy* expression in wildtype, *Npy*$^{+/-}$ and *Npy*$^{-/-}$ mice. Wildtype and *Npy*$^{+/-}$ mice show similar *Npy* expression, while *Npy*$^{-/-}$ mice do not show expression of NPY in stellate ganglia (*Figure 6G*). In our electrical stimulation study, we found that compared to *Npy*-expressing animals, *Npy*$^{-/-}$ mice exhibited muted heart rate responses at high (10 Hz and 15 Hz) but not low (1 Hz and 5 Hz) stimulation frequencies compared to mice expressing *Npy* (*Figure 6H, I*). To further support these findings, we investigated a similar experimental protocol in mice lacking the primary receptor mediating the cardiac effects of NPY (NPY1R; *Howell et al., 2005*). We compared *Npy1r* null mice to *Npy1r* expressing animals (*Npy1r*$^{-/+}$). Similar to mice with NPY-expressing neurons selectively depleted in stellate ganglia and mice lacking *Npy*, mice lacking the *Npy1r* exhibited muted heart rate responses at high stimulation frequencies (10 Hz and 15 Hz) compared to mice expressing *Npy1r*, while no differences were seen at low stimulation frequencies (1 Hz and 5 Hz) (*Figure 6J*). Taken together, these data suggest that NPY-expressing neurons (and by extension cardiac NPY-NPY1R signaling) are necessary at high sympathetic stress states (i.e. high stimulation frequencies), but not low sympathetic stress states (i.e. low stimulation frequencies). Such tiered control may explain how cardiac responses are graded in response to sympathoexcitation.

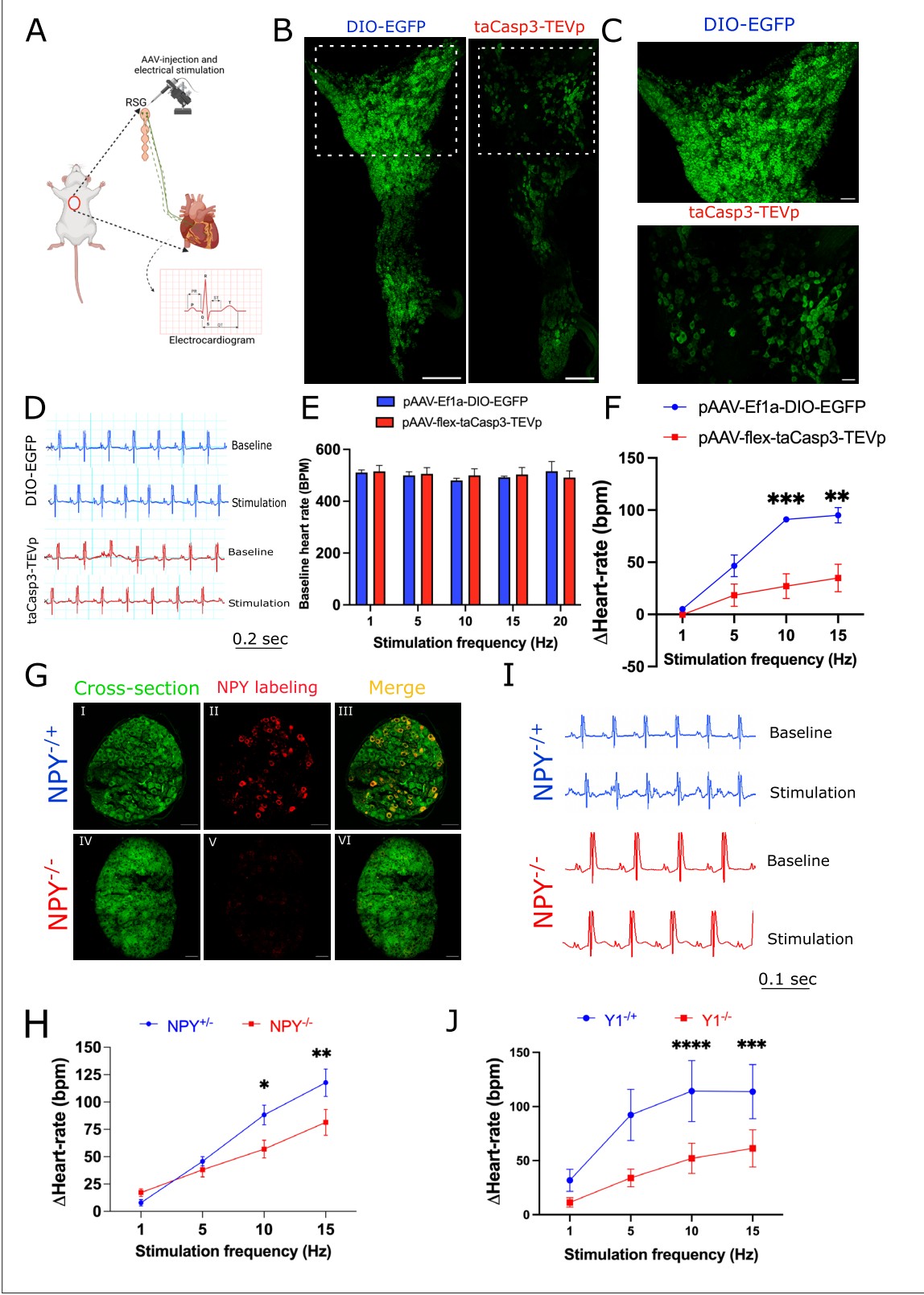

**Figure 6.** Cardiac sympathetic activation by stellate ganglionic neurons (SGNs) is mediated by NPY and receptor NPY-Y1R. (**A**) A schematic representation of electrical stimulation and chemogenetic ablation of *Npy*-expressing neurons in right stellate ganglia. (**B**) Whole mount right stellate ganglia prepared and stained with NPY antibody (green) from DIO-EGFP and taCasp3-TEVp injected mice. (**C**) High-magnification images obtained from DIO-EGFP and taCasp3-TEVp virus injected in right stellate ganglia. (**D**) The representative electrocardiogram snaps during right stellate ganglion

*Figure 6 continued on next page*

*Figure 6 continued*

stimulation (RSGS) from control and NPY-expressing neurons ablated mice. (**E**) Baseline heart rate for control and NPY neurons ablated mice at 1, 5, 10, and 15 Hz stimulation frequencies. (**F**) Heart rate change at 1, 5, 10, and 15 Hz stimulation frequencies in control vs. NPY neurons ablated mice. n=3 for control mice and n=5 for NPY neurons ablated mice. (**G**) Validation of Npy null mice. Right stellate ganglionic sections from *Npy$^{+/-}$* and *Npy$^{-/-}$* mice were stained with NPY antibody. (**G**) (I,IV) Autofluorescence to visualize tissue- architecture. (**G**) (II,V) NPY staining was observed only in tissue sections from *Npy$^{+/-}$* mice. (**H**) The representative electrocardiogram images from RSGS in *Npy$^{+/-}$* vs. *Npy$^{-/-}$* mice. (**I**) Change in heart rate at 1, 5, 10, and 15 Hz stimulation frequencies in *Npy$^{+/-}$* vs. *Npy$^{-/-}$* mice (n=8). (**J**) Heart rate change at 1, 5, 10, and 15 Hz stimulation frequencies in *Npy1r$^{+/-}$* vs. *Npy1r$^{-/-}$* mice (n=4). Data are shown as mean ± SEM. *p<0.05, **p<0.01, ***p<0.001, ****p<0.0001; Scale bars: 200 and 50 µm. The two-way ANOVA repeated measures followed by Sidak's multiple comparison test were used for the statistical analyses.

## Discussion

In the current work, we identified neuronal subpopulations in stellate ganglia that innervate myocardium, their transcriptomic profile, electrophysiological properties, morphometric parameters, and physiological role (*Figure 7*). The major findings of this study are; (**1**) three unique subtypes of stellate ganglion neurons innervate the heart, although the same subtypes may innervate other tissue beds; (**2**) although these three subtypes all synthesize catecholamines, the differ in NPY expression, such that neuronal subtypes with high *Npy* expression (i.e. NA1a subtype) are morphologically, transcriptionally, and electrophysiologically distinct cells; and (**3**) NPY release, presumably from NA1a neurons is required to achieve maximal cardiac sympatho-excitation. These findings may provide rationale supporting new therapeutic approaches that target specific neuronal subtypes in heart failure.

Stellate ganglionic neurons (SGN) innervate a variety of tissue beds and organs relying principally on cholinergic (Ach) or adrenergic (NE) neurotransmitters (*Furlan et al., 2016*). However, whether SGN subtypes are restricted to particular tissue beds is not understood. Our study identified significant overlap between subtypes that innervate the heart (NA1a, NA1b, and NA3) and the paw (NA1a, NA1b, NA1c, and NA3), although paw tissue has more neuronal subtypes than heart. These findings suggest that multiple SGN subtypes contribute to peripheral nervous system control of organs and tissues. These findings are consistent with those of *Furlan et al., 2016*, who used scRNAseq to explore SGNs innervating nipple- and pilo-erector muscles, finding that these neurons acquired cell type specification during organogenesis via the acquisition of unique combinations of growth factor receptors, and expression of ligands by nascent target tissues.

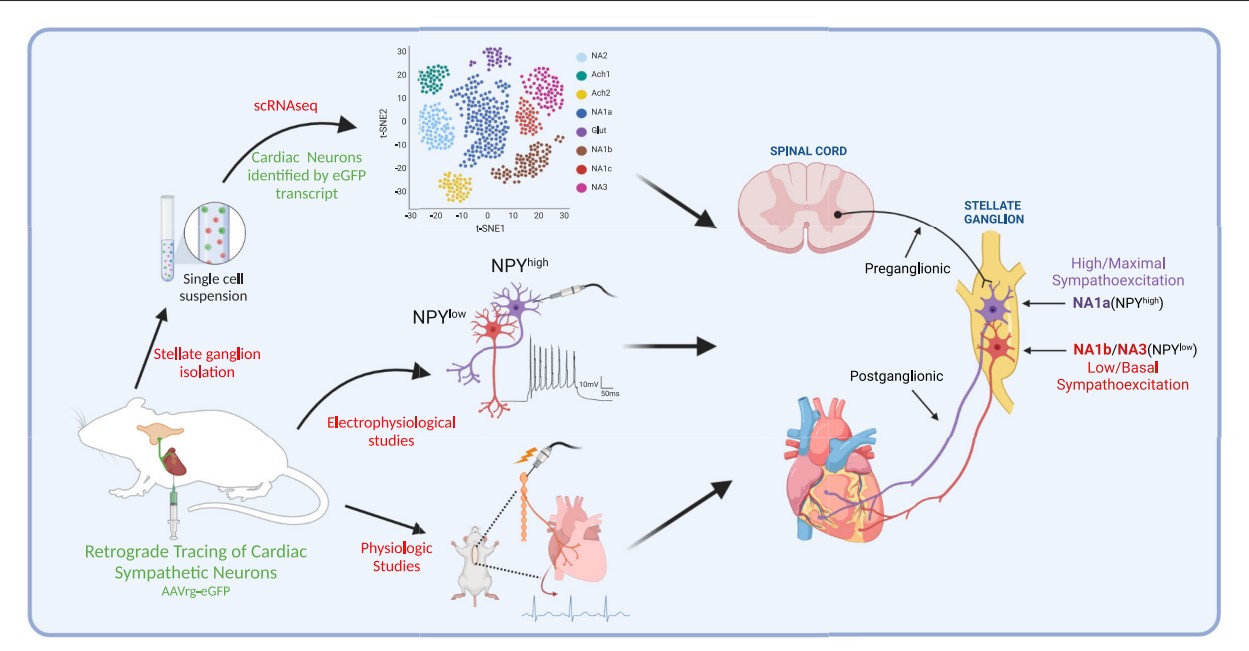

**Figure 7.** A schematic diagram of the approach taken in the current study. We identified and characterized cardiac innervating high- or low-*Npy* expressing neuronal subtypes that reside in the stellate ganglia. We identified intracardiac distribution of *Npy*-expressing cells, characterized their structural and electrophysiological properties, and determined the physiological function of *Npy* in relation to cardiac contractility.

The current study identified specific subsets of cardiac-innervating neurons in stellate ganglia distinguished by NPY expression. NPY, a key co-transmitter released from sympathetic nerve endings, is implicated in a diverse array of metabolic functions including temperature regulation, hunger and satiety, and cardiac adrenergic regulation (*Ajijola et al., 2020*; *Engström Ruud et al., 2020*; *Kalla et al., 2020*; *Tatemoto et al., 1982*; *Yan et al., 2021*). NPY receptors are expressed by various tissues (*Kuo et al., 2007*; *Marsh et al., 1998*; *Pedrazzini et al., 1998*), and in the heart, coupled with downstream signaling mechanisms that amplify cardiomyocyte calcium transients and increase diastolic calcium release from the sarcoplasmic reticulum (*Bryant and Hart, 1996*; *Pellieux et al., 2000*; *Protas et al., 2003*). Furthermore, the importance of NPY as a cardiac neurotransmitter is underscored by the recent finding that cardiac sympathetic blockade using beta-adrenergic receptor antagonists without NPY antagonists did not significantly raise ventricular fibrillation threshold, however, concomitant antagonism of both NPY and beta-receptor signaling did (*Kalla et al., 2020*). Additionally, low frequency sympathetic ganglion stimulation induced cardiac norepinephrine release, however, high frequency stimulation induced norepinephrine and NPY release into the myocardium from distal axonal projections from sympathetic neurons (*Kluge et al., 2021*). These data, interpreted in light of the new findings in this study, suggest that cardiac sympathetic regulation exists as a two-tiered system, where low-level cardiac sympathoexcitation is mediated by NPY-low or NPY-negative neurons, whereas high-level sympathoexcitation requires the recruitment of NPY expressing neurons.

A few findings in the current study support this notion. First, distinct neurons are high or low in NPY, suggesting that when robust cardiac sympathetic activation is required for the fight or flight responses, subsets of neurons rich in NPY are recruited. Second, mice with NPY ablated neurons in stellate ganglia are unable to reach maximal levels of cardiac chronotropic response during escalating sympathetic stimulation frequencies, unlike wildtype littermates. Given the profound actions of NPY as a potentiator of adrenergic signaling, our findings provide a basis for the prognostic role of elevated circulating or coronary sinus levels of NPY for identifying patients with severe sympathoexcitation for whom outcomes are dismal. Although, we have focused on NPY-expressing neurons, populations of neurons expressing other markers may also represent targets in disease states.

Of note, therapies that target the stellate ganglion, rather than systemic blockade, have shown significant benefit and are increasingly used clinically. Yet, these approaches target the entire stellate ganglion with potential off-target consequences for the other tissue beds and organs innervated by stellate ganglion neurons. Hence, specific targeting of neurons that innervate the heart and have a stronger influence in driving excessive chronic sympathetic signaling characteristic of chronic cardiac injury is needed. Coupling such cell-specific targeting with localized delivery of such agents to the stellate ganglion via a minimally invasive approach (for example used in concert with stellate ganglion block) offers a novel avenue to target sympathetic excess in chronic cardiac injury.

The current study has limitations. Although we carefully optimized tissue-specific labeling, we cannot completely exclude non-specific labeling of a few cells in stellate ganglia. However, our IHC studies validated the subpopulations identified by scRNAseq.

## Conclusions

In summary, we have identified the distinct subtypes of SGNs that innervate the heart and shown that these subtypes exert functional control based on expression of *Npy*. We present evidence that the absence of NPY-expressing subpopulations in the stellate ganglia results in the inability to achieve maximal sympathoexcitation. These findings expand our understanding of cardiac sympathetic control mechanisms and suggests that cell-specific targeting of sympathetic neurons may offer new therapies for heart failure.

## Methods

### Mouse strains

Animal experiments complied with institutional guidelines and ethical regulations. The study (protocol number: 18–048) was approved by the UCLA institutional Animal Care and Use Committee. The ethical approvals for the use of Adeno-associated virus vectors (AAVs) were provided by the Institutional Biosafety Committee (IBC), UCLA. Adult male mice, aged between 8 and 10 weeks, were housed according to the standard laboratory conditions (12 hr light/dark) with ad libitum access to

food and water. Mice strains C57BL/6 J (000664), *Npy*-hrGFP (006417), and *Npy*-IRES-Cre (027851) were purchased from the Jackson Laboratory. *Npy* and *Npy1r* knockout mice were received as a generous gift from Dr. Herbert Herzog's laboratory in Garvan Institute of Medical Research, Australia.

## Retrograde neuronal tracing experiments

We used C57BL/6 J and *Npy*-hrGFP mice strains for tracing neurons in the stellate ganglion. We injected 10 µl of each retrograde tracer AAV or cholera toxin-B (CTB) from heart. AAVs serotype AAV2/retro-CAG-flex-tdTomato (1e+12 gc/ml) and AAV2/retro-CAG-flex-GFP (1e+12 gc/ml) were procured from Viral vector core, Boston Children's Hospital. The cholera toxin subunit-B conjugated; Alexa Fluor-488 (C-34775), Alexa Fluor-555; (C-34776), Alexa Fluor-647 (C-34778) were procured from Molecular Probes.

## Cardiac injections of CTB and AAVs

C57BL/6 J and *Npy*-hrGFP transgenic mice were given carprofen (5 mg kg$^{-1}$, s.c.), cefazolin (Hikma Farmaceutica, #0143-9924-90) and buprenorphine (0.05 mg kg$^{-1}$, s.c.) 1 hr before surgery. Animals were anesthetized, intubated, and mechanically ventilated. A left lateral thoracotomy was performed at fourth intercostal space, pericardium opened, and heart exposed. Ten µl of either CTB conjugate to Alexa Fluor-488, or –555, or 647 (0.2%) or AAV conjugated to GFP or tdTomato (1e+12 gc/ml) was injected intramyocardially in the heart with a 31-gauge needle. The surgical wounds were closed with 6–0 sutures. Animals were sacrificed, perfused with 4% paraformaldehyde (4% PFA), and right and left stellate ganglion tissues were harvested 4 days of post-surgery.

## Stellate ganglia dissociation for single-cell analyses

Bilateral stellate ganglia in C57BL/6 mice were identified, isolated, and collected in artificial cerebro-spinal fluid (ACSF; *Figure 1A*). Next, 16 left and right stellate ganglia combined were incubated for an hour at 37 °C in a digestion solution prepared with 500 µl TrypLE Express (ThermoFisher Scientific, Waltham, Massachusetts), 2000 µl Papain solution (Worthington Biochemical Corporation, Lakewood, New Jersey; 25 units/mL in ACSF), 100 µl Collagenase-Dispase (Millipore Sigma, Burlington, Massachusetts; 20 mg/mL in ACSF) and 270 µl DNAse I (Worthington Biochemical Corporation, Lakewood, New Jersey; 200 units/mL in ACSF). After an hour, the cells were carefully triturated with fire-blown Pasteur pipettes every thirty minutes. Following the second trituration, 100 µl fresh Collagenase-Dispase (20 mg/mL in ACSF) was added to the solution and cells were incubated for another hour at 37 °C, triturating every thirty minutes. Next, the suspension was filtered through a 40 µm filter (Thermo Fisher Scientific, Waltham, Massachusetts) and ACSF was added to halt enzymatic digestion. The suspension was spun down at 100 *g* for 4 min at room temperature, and the pellet collected and resuspended in 500 µL ACSF and 500 µL supplemented Neurobasal-A medium (Thermo Fisher Scientific, Waltham, Massachusetts; 250 µL B27, 250 µL Penicillin-streptomycin, 31.3 µL L-Glutamine). The cell suspension was transferred onto a density gradient (Millipore Sigma, Burlington, Massachusetts) and centrifuged at 100 *g* for 10 min at room temperature. The supernatant was carefully removed until the solution was concentrated to 500 µL. Singlecell RNA sequencing used microfluidic capture-based encapsulation, barcoding, and library preparation 10 x Genomics Chromium scRNAseq system (10 x Genomics, Pleasanton, California). Cells were loaded into a Chromium Chip B along with partitioning oil, reverse transcription reagents, and a mix of hydrogel beads containing 3,500,000 unique 10 X Barcodes. Paired end sequencing was performed on a Novaseq S4 system, using the v3 Illumina platform. Coverage depth was 20 K per cell, and the read length was 2x50. Analysis including demultiplexing, reference-based mapping (GRCm38.98), and UMI identification, was performed according to the 10 X Cell Ranger pipeline. We modified the reference by adding in the sequences for GFP to detect their transcripts in cells.

## scRNAseq data quality control, normalization, and integration

To be included in the dataset, cells had to express more than 500 but fewer than 6000 genes. Dataset requirements call for fewer than 20,000 total UMIs detected, and less than 15% of their RNA content to be from mitochondria. The following scRNAseq analysis was performed in the Seurat v3 R package. We then split the data by sample, normalized the data, found the top 2000 highly variable features, and integrated the samples together to remove any batch or sample-specific effects. We then scaled

the data, while regressing out effects from the number of genes and mitochondrial percent. We performed dimensionality reduction using principal component analysis (PCA) with 30 PCs. We computed the nearest neighbor embedding on 20 principal components (PCs), and further reduced the dimensionality to two dimensions with UMAP for visualization. We performed Louvain community detection, to separate cell types into clusters. We found cluster markers by the Wilcoxon test (FDR < 0.05) and identified cell types from these markers. We identified endothelial cells, fibroblasts, macrophages, microglia, neurons, satellite glia, Schwann cells, smooth muscle cells, T cells, and VLMC. We classified the neuronal population based on the expression of *Snap25*, *Syt1*, *Pcsk1n*, and *Prph*, separating these neurons out for further analysis.

## Single-cell clustering of neuronal subpopulation

To identify the neuronal subtypes from neuronal populations, we processed scRNAseq data as mentioned in the supplementary method section 3. First, we integrated the data as described, reduced the dimensionality with PCA on 30 PCs, found the nearest neighbor embedding, and determined the 8 subclusters with the Louvain community detection algorithm. We identified marker genes for each cluster by the Wilcoxon test (FDR < 0.05). These neuron subclusters were annotated according to previously defined neuronal cell subtypes (*Furlan et al., 2016*). We identified 5 neuro adrenergic clusters (NA1a, NA1b, NA1c, NA2, NA3), 2 acetylcholine (Ach1, Ach2), and 1 glutamatergic subtype (Figure S2 and S3). We identified cardiac and paw innervating neurons based on the expression of GFP and tdTomato transcripts respectively. We determined *Npy* (low, medium, and high) neuron subpopulations based on their presence in the bottom 30%, middle 40%, or top 30% quantile. We identified *Npy*-associated genes in the cardiac (NA1a, NA1b, NA3) and paw (NA1a, NA1b, NA1c, NA3) innervating subclusters. For each gene, to eliminate the effects of high zero content, we identified cells expressing at least one transcript of both *Npy* and the other gene of interest. We then computed the Pearson correlation coefficient, and corrected p-values based on the Benjamini-Hochberg procedure. Significant associations (FDR < 0.05) were identified for further analysis. To visualize the association of receptors, transcription factors, neuropeptides, and ion channels, we split the neurons into the 3 categories of *Npy* expression, low, medium, and high. The bottom 30% based on *Npy* were categorized as low/negative, the middle 40% were categorized as medium, and the top 30% were categorized as high *Npy* expressions. We then visualized the relative expression of each gene with heatmaps using the seaborn python package.

## Pathway enrichment analysis

We annotated biological pathways enriched with the *Npy* associated genes. We identified these pathways based on the negative, positive, and combined *Npy* associated genes. Pathways were determined based on the hypergeometric test with corrected p-values based on the Benjamini-Hochberg procedure. Significant pathways were those significantly enriched for *Npy* associated genes (FDR < 0.05). Pathways from the Biocarta, Hallmark, Kegg, and Reactome databases were considered.

## Dissection of thorax and isolation of stellate ganglia and heart tissues

Mice strains C57BL/6 J, *Npy*-hrGFP, and *Npy*-IRES-Cre were anesthetized with 4% isoflurane and euthanized by cervical dislocation. Mice were first perfused with 50 mL ice-cold 0.01 M phosphate buffer saline (PBS) (10 mM potassium phosphate, pH 7.4 and 150 mM NaCl) containing heparin followed by 50 mL freshly prepared, ice-cold 4% paraformaldehyde in PBS. Once the animal was fixed, tissues (left and right stellate ganglia and heart) were isolated, and fixed in 4% PFA overnight at 4°C, washed, and stored at 4°C. Heart tissues from AAVrg-CAG-GFP-WPRE injected mice collected, sectioned, and stained for rabbit PGP9.5 (Abcam, #ab108986, 1:400 dilution) and goat anti-cardiac troponin I to confirm the retrograde labeling of cardiac innervating neurons in the stellate ganglia.

## Cryo-sectioning and immunofluorescence tissue labeling

Bilateral stellate ganglion tissues from C57BL6/J, *Npy*-hrGFP, and *Npy* knockout mice were isolated and transferred into 24-well plates containing phosphate buffered saline (PBS; 10 mM potassium phosphate, pH 7.4 and 150 mM NaCl)-NaN3 (0.02%). Tissues were then replaced with 4% w/v paraformaldehyde in PBS for 24 hr of fixation. Tissues were removed from 24-well plates, washed with PBS, and kept in 30% sucrose (w/v) before embedding into cryo-medium OCT compound (Tissue-Tek,

Sakura, Netherlands). OCT embedded tissues were frozen and stored at –80 °C. Tissue blocks were then sliced into 25–30 µm thick sections on a cryostat (Leica Jung Frigocut 2000, Wetzlar, Germany) at –25 °C and collected in PBS containing 0.1% sodium azide. The samples were then permeabilized in 0.1% Tween-20 (Sigma-Aldrich) in PBS (0.1% PBS-T) for 10 min and washed three times with PBS for 5 min each. The samples were then blocked in 10% v/v normal donkey serum (NDS) (D9663, MilliporeSigma Corporate) in 0.2% PBS-T for 20 min and washed with PBS for 5 min. The primary antisera incubation was carried out either at 4°C overnight or 3 hr at room temperature (RT) in 0.2% PBS-T containing 5% NDS. The primary antibodies were: sheep anti-TH (Millipore, #ab1542, 1:200), rabbit anti-CNTN5 (Thermo Fisher Scientific, #PA5-59486, 1:30), rabbit anti-NPY (Cell Signaling Technology, #11976 S, 1:200), goat anti-NPY (Novus Biologicals, #NBP1-46535, 1:200), rabbit anti-VIP (Thermo Fisher Scientific, #PA5-78224, 1:100), rabbit anti-KCNT1/SLACK[N3/26] (Abcam, #ab94578, 1:50), goat anti-NGFR (R&D System, #AF1157, 1:100), goat anti-vimentin (Millipore Sigma, #V4630, 1:400), rabbit anti-calmodulin (Novus Biologicals, #NBP2-67413, 1:100), rabbit anti-SCTR (Thermo Fisher Scientific, #PA5-76698, 1:25), rat anti-SST (Millipore-Sigma, #MAB354, 1:50), goat anti-FABP7 (R&D system, #AF3166, 1:100), rabbit anti-CASZ1 (Thermo Fisher Scientific, #PA5-55800, 1:100), goat anti-cardiac troponin I (Abcam, #ab56357, 1:200), and rabbit anti-PGP9.5 (Abcam, #ab108986, 1:400) used for the specific key gene markers. Samples were then washed six times in PBS for 5 min each and incubated with secondary antibodies (1:200) in 0.2% PBS-T containing 5% NDS for 1 hr at RT. Secondary antibodies used for the detection were: donkey conjugated Cy3 IgG (H+L) anti-rabbit (Jackson ImmunoResearch, #711-165-152), anti-sheep (Jackson ImmunoResearch, #713-165-147), anti-goat (Jackson ImmunoResearch, #705-165-147), Alexa Fluor-488 IgG (H+L) anti-rabbit (Jackson ImmunoResearch, #711-545-152), anti-sheep (Jackson ImmunoResearch, #711-165-152); Alexa Fluor-647 IgG (H+L) anti-goat (Jackson ImmunoResearch, #705-605-147), and anti-rat (Jackson ImmunoResearch, #712-605-153). Immediately after secondary antibodies incubation, tissue-sections were washed with 0.01 M PBS six times for 5 min each. The samples were mounted onto SuperFrost Plus slides (Thermo Fisher Scientific, #12-550-15) using the anti-fade reagent, n-propyl gallate (Sigma-Aldrich) in 100 mM phosphate buffer (pH 7.2).

For pig and human stellate ganglia samples (*Figure 2*), paraffin-embedded ~5-µm-thick sections were placed on slides using the microtome. Antigen retrieval was carried out using the antigen unmasking solution (ab93680) in a pre-heated chamber (95 °C) for 20 min. Samples were treated with autofluorescence quenching solution (TrueBlack Lipofusein, #23007, Biotium) before blocking with a buffer containing 10% NDS in 0.2% PBS-T, and then primary and secondary antibody incubations carried out as mentioned above. Tissues were washed several times in 0.01 M PBS before being mounted on the slides using RIMS (refractive index matching solution).

## Confocal microscopic imaging

High-fluorescent images were acquired on a confocal laser scanning microscope LSM 880 Upright (Carl-Zeiss, Germany) in fluorescence and bright-field illumination modes. The laser lines used were 405 nm (diode [405-30]) for excitation of DAPI, 488 nm (argon) for Alexa-488, 561 nm (DPSS [561-10]) for Cy3, and 633 nm (HeNe) for Alexa-647. Samples were visualized through Plan-apochromat 10 x/0.45 air, Plan-Apochromat 20 x/0.8 air, and Plan-Apochromat 63 x/1.4 oil objectives. For imaging the whole-mount tissues, we used z-stack optical sectioning and tile-scan image overview mode to generate the composite images.

## Image processing, quantification, and data analysis

Cardiac and other tissue innervating neuronal subpopulations in the stellate ganglia were processed and quantified using ZEN-black (ZEISS), NIH ImageJ, and Imaris softwares. To validate the cardiac innervating neuronal subtypes, we stained stellate ganglia sections, collected from C57BL/6 J and *Npy*-hrGFP mice labeled retrogradely from heart and other tissues using either CTB-647 or –555. To count the NPY + cardiac innervating neurons, first, we counted only CTB + cells in *Npy*-hrGFP mice injected with CTB-555 in heart (Apex and base) and paw using the analyze particles function tool in the ImageJ and then counted CTB + cells colocalized with NPY + cells. This method allowed us to quantify the NPY +and NPY- neurons that innervate different regions of the heart (*Figure 4F, G, H and J*).

## Neuronal electrophysiology

For intracellular recording from intact preparations of whole right stellate ganglia using sharp microelectrodes, adult male *Npy*-hrGFP mice (n=11) were sacrificed under deep isoflurane (5%) anesthesia by cervical dislocation and exsanguination. The thorax was removed and placed in the ice-cold physiological salt solution (PSS) containing, in mM: 121 NaCl, 5.9 KCl, 1.2 $NaH_2PO_4$, 1.2 $MgCl_2$, 25 $NaHCO_3$, 2 $CaCl_2$, 8 D-glucose; pH 7.4 maintained by 95% $O_2$-5% $CO_2$ aeration. Stellate ganglia were isolated from the paravertebral chain at the origin of the first rib, and the overlying connective tissue was pinned to the SylGard (Dow Corning) floor of a glass bottom petri dish. Isolated ganglia were superfused (6–7 ml/min) continuously with PSS maintained at 32–35°C with a thermostatically controlled heater. Stellate neurons, either positive or negative for GFP expression as identified under epifluorescence, were impaled with borosilicate-glass microelectrodes filled with 2 M KCl +2% Neurobiotin (Vector Labs). Membrane voltage was recorded in a current clamp configuration using a Multiclamp 700B amplifier connected to a Digidata 1550B data acquisition system and personal computer with pCLAMP 10 software (Molecular Devices, CA) for data acquisition and analysis.

## Virus administration and electrical stimulation of right stellate ganglia

The pAAV-flex-taCasp3-TEVp (AAV5) was a gift from Niroa Shah and Jim Wells (Addgene plasmids # 45580) and pAAV-Ef1α-DIO-EGFP (AAV5) was a gift from Bryan Roth (Addgene plasmids # 27056). *Npy*-IRES-Cre (n=8), *Npy* knockout (*Npy*$^{-/-}$, n=8; *Npy*$^{-/+}$, n=8), and Npy1r knockout (Npy1r$^{-/-}$, n=4; Npy1r$^{-/+}$, n=4) mice were anesthetized with isoflurane (induction at 4%, maintenance at 2%, inhalation) (Isoflurane Vaporizer, SOMNI Scientific). Two needle electrodes were inserted subcutaneously into the right forearm and the left hindleg to obtain lead II electrocardiograms (Grass RPS 107 Regulated Power Supply, Grass Instrument Co.; Grass P511 AC Amplifier, Natus Neurology). With the use of a dissecting microscope (SZ-CTV Olympus Dissecting Microscope; KL 2500 LED Illuminator, Schott), a midline neck incision was performed to intubated, and mechanically ventilated mice at 150 bpm (SAR-830/P Ventilator, CWE, Inc). A left thoracotomy was performed at the second intercostal space to expose the right stellate ganglia located along the sympathetic chain near the first rib. Mice were injected with either DIO-EGFP or DIO-EGFP +taCasp3 TEVp (3 µl; 1e+13 vg/ml) into the right stellate ganglia using Programmable Nanoliter Injector (Nanoject III, Drummond Scientific Company). The viral load was released into the stellate ganglia with a rate of 20 nl per second. Five minutes later, the glass capillary pipette was pulled out slowly from right stellate ganglia. Mice were recovered on heating pads. AAVs were allowed to be incubated for three weeks in mice before electrical stimulation of right stellate ganglia was performed.

All electrode stimulations (Concentric Bipolar Electrode, FHC) were performed with repeating, monophasic pulses for 10 s at a constant duration of 10ms, output amplitude of 10mA, for a period of 10 s, while varying the stimulation frequencies by 1 Hz, 5 Hz, 10 Hz, 15 Hz, and 20 Hz (Grass S88x Stimulator, Grass-Technologies Astro-Med, Inc; Photoelectric Stimulus Isolation Unit, Grass Instrument Co.; PowerLab/16SP, AD Instruments Ltd). A period of 10 min between the stimulations allowed the heart rate to return to baseline. Electrocardiograms were recorded using LabChart 7 and analyzed with LabChart7 and Spike2 7.11 programs. Right stellate ganglia were collected, stained, and imaged for NPY expression after the electrical stimulation.

## Statistical analysis

Microsoft Excel and GraphPad Prism 9.0.1 were used for data handling, analysis, and graphs. Sample sizes and statistical tests performed are indicated in the legend for each figure. Normality of data was assessed using the Shapiro-Wilk or Kolmogorov–Smirnov test. Normally distributed data were compared through a Welch's t-test or ANOVA, while data not normally distributed were analyzed through a Mann-Whitney test. Legend for statistical significance: *$p \leq 0.05$, **$p \leq 0.01$, ***$p \leq 0.001$, ****$p \leq 0.0001$. All data are shown as individual data points.

## Software packages

scRNAseq processing: R, cell ranger, Seurat v3. Visualization: python, scaNpy, pandas, numpy, matplotlib, seaborn. Npy association: python scipy.

## Acknowledgements

The authors wish to thank Timothy Davis, Zhiwei Li, Amiksha Gandhi MS, and Pratap Meera PhD for technical assistance; One Legacy for human stellate ganglia; and Pradeep Rajendran, Felix Schweizer PhD, Emilie Marcus PhD, and Kalyanam Shivkumar MD PhD for helpful comments on the studies and manuscript. This study is supported by postdoctoral fellowship (906065) to SS from the American Heart Association and NIH OD DP2HL142045 and R01HL162717 to OAA.

## Additional information

### Competing interests

Patrick Y Jay: is affiliated with Alnylam Pharmaceuticals. Olujimi A Ajijola: Reviewing editor, *eLife*. The other authors declare that no competing interests exist.

### Funding

| Funder | Grant reference number | Author |
| --- | --- | --- |
| NIH Office of the Director | DP2HL142045 | Olujimi A Ajijola |
| NHLBI Division of Intramural Research | R01HL162717 | Olujimi A Ajijola |
| American Heart Association | | Olujimi A Ajijola |

The funders had no role in study design, data collection and interpretation, or the decision to submit the work for publication.

### Author contributions

Sachin Sharma, Software, Formal analysis, Supervision, Validation, Investigation, Visualization, Methodology, Writing - original draft, Project administration, Writing – review and editing; Russell Littman, Data curation, Software, Formal analysis, Writing - original draft; John D Tompkins, Data curation, Formal analysis, Methodology; Douglas Arneson, Data curation, Formal analysis; Jaime Contreras, Data curation, Methodology; Al-Hassan Dajani, Kaitlyn Ang, Software, Methodology; Amit Tsanhani, Software; Xin Sun, Formal analysis; Patrick Y Jay, Methodology, Writing – review and editing; Herbert Herzog, Knockout mice NPY-KO and NPY Y1-KO were procured from his laboratory; Xia Yang, Supervision, Writing – review and editing; Olujimi A Ajijola, Conceptualization, Resources, Supervision, Writing – review and editing

### Author ORCIDs

Sachin Sharma ![ORCID] http://orcid.org/0000-0002-6776-1061
John D Tompkins ![ORCID] http://orcid.org/0000-0001-9496-7930
Xin Sun ![ORCID] http://orcid.org/0000-0001-8387-4966
Herbert Herzog ![ORCID] http://orcid.org/0000-0002-1713-1029
Olujimi A Ajijola ![ORCID] http://orcid.org/0000-0001-6197-7593

### Ethics

Animal experiments complied with institutional guidelines and ethical regulations, and the study protocol was approved by the UCLA institutional Animal Care and Use Committee. (protocol number: 18-048).

### Decision letter and Author response

Decision letter https://doi.org/10.7554/eLife.86295.sa1
Author response https://doi.org/10.7554/eLife.86295.sa2

## Additional files

### Supplementary files
- Supplementary file 1. RNA Sequencing Information.
- Supplementary file 2. Additional Data on Sequenced Genes.
- MDAR checklist

### Data availability
Data related to single-cell RNA seq analysis generated from this manuscript are available from the GEO database (GSE231924).

The following dataset was generated:

| Author(s) | Year | Dataset title | Dataset URL | Database and Identifier |
|---|---|---|---|---|
| Ajijola O, Sharma S, Littman R | 2023 | Tiered Sympathetic Control of Cardiac Function Revealed by Viral Tracing and Single Cell Transcriptome Profiling | https://www.ncbi.nlm.nih.gov/geo/query/acc.cgi?acc=GSE231924 | NCBI Gene Expression Omnibus, GSE231924 |

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
