## [Editor Report]

This is a key paper reporting the identification of sympathetic ganglion subtypes with differing neuropeptide Y levels, gene expression, electrophysiological properties, and roles in cardiac sympathetic excitation of the heart. Molecular and cellular level findings are validated by their physiological effects. This is a research programme of fundamental and potentially therapeutic importance.

---

## [Decision Letter]

**Decision letter after peer review:**

Thank you for submitting your article "Tiered Sympathetic Control of Cardiac Function Revealed by Viral Tracing and Single Cell Transcriptome Profiling" for consideration by *eLife*. Your article has been reviewed by 3 peer reviewers, one of whom is a member of our Board of Reviewing Editors, and the evaluation has been overseen by Mone Zaidi as the Senior Editor. The following individual involved in the review of your submission has agreed to reveal their identity: Christian Meyer (Reviewer #3).

We unanimously congratulate you on an important contribution that will lead to new therapeutic approaches directed at cardiac neural control. We make comments intended to be constructive in revision of the paper.

Essential revisions:

Reviewer 1 has no requests to make.

A) Major clarification requested by reviewer 2:

Viral administration and physiological studies were performed on the right, while RNA sequencing was done from the right and left stellate ganglion. As there are physiological lateral differences between the effects of the left and right stellate ganglion, it would be useful to thoroughly report which side was used for which experiment throughout the manuscript and to discuss whether any lateral differences are relevant for the obtained results and conclusions.

B) Minor suggestions from reviewer 2:

Introduction:

- The abbreviation PGN is not needed.

- Skin and paw: please provide a reference. Also, please specify in the introduction whether the data are from human or animal studies (or both).

Methods

- Injection into the subepicardium in mice: how many cell layers is the subepicardium defined in mice? Can the authors differentiate myocardium from subepicardium in mice? Please elucidate how many injections were performed per heart and in which locations. How was the left posterior wall targeted, where innervation is more dense?

- In suppl Figure 5A, GFP seems to be located in many cardiomyocytes additionally to nerve fibers. GFP+ cardiomyocytes are also present in S5 C+D. How do the authors explain this with their technique?

- How many stellate ganglia were used for RNAseq?

Results

- Figure 2: the colour scheme makes it very hard to understand the figure and it seems redundant to include Figure 2 when Figure S8 has more information. Also, in Figure S8, it's hard to distinguish high and low NPY subtypes. I suggest using single images in grayscale for an unbiased view.

- Also: Can the authors comment on the position of these cells within human (porcine) stellate ganglion?

- What is the rationale to use heterozygous animals and not wild type as controls for NPY and Y1R mice? do these animals exhibit any relevant phenotypes?

- "NPY-/- mice showed merely any expression of NPY in stellate ganglia (Figure 6G)" How do the authors explain that they still detect NPY? Is the antibody specific? Are any parts of the protein still being expressed?

- What is the rationale to use heterozygous animals and not the wild type used as controls for NPY and Y1R mice?

Limitations

- The limitation sections start with "first" and do not continue.

C) Minor suggestions from reviewer 3

1. Why was Neuropeptide Y focused as a potential "target"? Which data beyond previous publications were key for this decision? Could the authors speculate and/or present additional data on potential other promising candidates? Are there additional data available with respect to age and sex? At least please discuss.

2. The abbreviation NPY is not explained in the summary.

3. In the last sentence of the first paragraph of the discussion the authors mention that

"These findings lay the groundwork for new therapeutic approaches that target specific neuronal subtypes in heart failure."

This is correct but it might be helpful to be rephrased since no in-depth studies from heart failure models are presented in the present manuscript. Therefore, it is not known whether and how the present innovative findings hold true with aging and in diseased animals and humans.

4. This/point 3 is also important since heart failure goes along with profound changes within the cardiac neural control (as partly outlined by the authors in the introduction). In this context, the concept of transdifferentiation during the development and progression of heart failure might come into play.

5. Also it would be of interest in more detail to the reader and community to get additional insights on how the here presented cell types and insights are related to "classical neurotransmitters", other neuropeptides, and how the neuronal subtypes might interfere with other cells/cell types like Schwann cells, microglia, satellite glia.

---

## [Author Response]

Essential revisions:A) Major clarification requested by reviewer 2:Viral administration and physiological studies were performed on the right, while RNA sequencing was done from the right and left stellate ganglion. As there are physiological lateral differences between the effects of the left and right stellate ganglion, it would be useful to thoroughly report which side was used for which experiment throughout the manuscript and to discuss whether any lateral differences are relevant for the obtained results and conclusions.

We appreciate the reviewer’s point. Indeed, there are known physiologic differences between the right and left stellate ganglion (RSG & LSG, respectively). The RSG innervates the SA node and anterior ventricles, while the left innervates the posterior aspect of the ventricles (PMID: 21917265, PMID: 28087519, PMID: 23893167, PMID: 22465457). While we performed scRNAseq and immunohistochemistry on both stellate ganglia we performed physiologic studies focused on heart rate by manipulating the RSG (as the LSG would not increase heart rate). For scRNAseq analysis, immunohistochemistry validation, and morphometric analysis, we performed experiments in RSG and LSG while neuronal electrophysiology and virus administration and RSG stim experiments were performed only on the right stellate ganglia. In the revised manuscript, we added text at line numbers 608, 611, 692, 699, 765, 830, 867, 879, 895, 901, and 947 about specific stellate ganglia used for specific experiments. Additionally, legends for figures 1, 4, and 6 have been revised as well.

B) Minor suggestions from reviewer 2:Introduction:- The abbreviation PGN is not needed.

Thank you for highlighting this. We have removed the abbreviation PGN from the introduction.

- Skin and paw: please provide a reference. Also, please specify in the introduction whether the data are from human or animal studies (or both).

We have added references in line 84 for the skin (Taniguchi et al., 1994) and paw (Li and Dahlström, 2008). We have specified that data are obtained specifically from animal studies except for neuronal subtypes identification in human and porcine stellate ganglionic tissues (Figure 2).

Methods- Injection into the subepicardium in mice: how many cell layers is the subepicardium defined in mice? Can the authors differentiate myocardium from subepicardium in mice? Please elucidate how many injections were performed per heart and in which locations. How was the left posterior wall targeted, where innervation is more dense?

The mouse heart, histologically, has three defined layers known as epicardium, midmyocardium, and endocardium. In the field, subepicardial is used interchangeably with intramyocardial. Given how thin the wall of the mouse heart is, we used the term subepicardial to simply connote that it was injected into the wall of the heart beneath the epicardium. To clarify this, we changed the term to intramyocardial in the manuscript.

We used 6mm, 30-gauge insulin syringes to inject 10ul of retrograde tracer intramyocardially. We performed only one injection per site only either at the apex or base of the heart, and each heart had 2-3 injections. These have been added to the methods section line 94 and 606.

- In suppl Figure 5A, GFP seems to be located in many cardiomyocytes additionally to nerve fibers. GFP+ cardiomyocytes are also present in S5 C+D. How do the authors explain this with their technique?

This is likely due to the fact the virus we used rAAV-CAG-GFP was taken up by cardiomyocytes at the local injection site leading to GFP expression in cardiomyocytes. This was helpful as it confirmed that the virus was indeed injected into the heart, as well as the sites of injection. The presence of GFP in nerve fibers is due to uptake and retrograde transport by sympathetic nerve endings to the soma, transcription in the soma, and antegrade transport of GFP from the soma to sympathetic nerves in the heart. This was the expected finding.

- How many stellate ganglia were used for RNAseq?

We have used 16 total stellate ganglia including both the sides right and left from total of 8 mice for scRNAseq analysis. We added into method section at line 611.

Results- Figure 2: the colour scheme makes it very hard to understand the figure and it seems redundant to include Figure 2 when Figure S8 has more information. Also, in Figure S8, it's hard to distinguish high and low NPY subtypes. I suggest using single images in grayscale for an unbiased view.

We appreciate the reviewer for bringing this to our attention. We have replaced figure 2 with figure S8. However, we report the colocalization of specific markers within the same cell type and it would be difficult to distinguish subtypes using a grayscale image. The tissue sections used were from the cardiac-projecting region of the stellate ganglia.

- What is the rationale to use heterozygous animals and not wild type as controls for NPY and Y1R mice? do these animals exhibit any relevant phenotypes?

We did not observe any relevant phenotypic differences between wild-type and heterozygous animals from NPY KO and Y1R mice. Our stimulation studies included wildtype and heterozygous mice because we wanted to differentiate between having NPY (i.e. WT and NPY^+/-^) and completely lacking NPY (i.e. NPY^-/-^). We compared and did not find differences between WT and NPY^+/-^ animals, so these groups were combined. This is explained in the methods section. Our immunostaining for NPY in the stellate ganglia from wild-type and heterozygous mice also did not show differences indicating that there was no significant difference in NPY expression. However, NPY^-/-^ mice completely lacked NPY expression in stellate ganglia as depicted in Figure 6G.

- "NPY-/- mice showed merely any expression of NPY in stellate ganglia (Figure 6G)" How do the authors explain that they still detect NPY? Is the antibody specific? Are any parts of the protein still being expressed?

This is a typographical error, for which we apologize. NPY-/- ganglia do not show expression of NPY. We immunostained stellate ganglia from several mice and the results were consistently negative, however, for heterozygous and wild-type mice, immunostaining detects robust NPY expression.

Limitations- The limitation sections start with "first" and do not continue.

We removed the word “first” from the paragraph.

C) Minor suggestions from reviewer 31. Why was Neuropeptide Y focused as a potential "target"? Which data beyond previous publications were key for this decision? Could the authors speculate and/or present additional data on potential other promising candidates? Are there additional data available with respect to age and sex? At least please discuss.

NPY is a co-transmitter released by sympathetic nerve terminals along with norepinephrine to dynamically modulate the cardiac function (Tan et al., 2018; Zhu et al., 2016). Elevated circulating NPY levels are associated with adverse outcomes in chronic heart failure patients and people suffering from arrhythmia (Ajijola et al., 2020; Kalla et al., 2020). All of these previous studies suggest that NPY is important in the pathogenesis of sympathetic neuronal dysfunction. The rationale for focusing on NPY is that we found it striking that distinct groups of neurons expressing NPY vs. not expressing NPY innervated the heart. Given what we know about NPY, we were immediately interested in this population as they may be a target in chronic heart failure. We have added this into the paper (Line 101).

The comment about other potential candidates is an important one. Since we did not characterize any other populations, we avoided speculating about other potential targets, however, they include Galanin-expressing neurons. We have revised the discussion to clarify that although we have focused on NPY-expressing neurons, populations of neurons expressing other markers may also represent targets in disease states (line 386).

We did not specifically investigate age and sex-related differences in this population of neurons and agree that this interesting question should be the objective of future investigations. We could not find any cardiovascular literature studying age and sex-related differences in NPY-expressing neurons.

2. The abbreviation NPY is not explained in the summary.

We have explained the abbreviation neuropeptide Y (NPY) in the summary.

3. In the last sentence of the first paragraph of the discussion the authors mention that"These findings lay the groundwork for new therapeutic approaches that target specific neuronal subtypes in heart failure."This is correct but it might be helpful to be rephrased since no in-depth studies from heart failure models are presented in the present manuscript. Therefore, it is not known whether and how the present innovative findings hold true with aging and in diseased animals and humans.

As per reviewers’ suggestion, this sentence has been rephrased in the manuscript (line 339).

4. This/point 3 is also important since heart failure goes along with profound changes within the cardiac neural control (as partly outlined by the authors in the introduction). In this context, the concept of transdifferentiation during the development and progression of heart failure might come into play.

Thank you for the informative comment. Transdifferentiation of neurons within the neuronal subpopulation is a common phenomenon and has been reported (PMID: 20051627) previously. We completely agree and will investigate this in future work.

5. Also it would be of interest in more detail to the reader and community to get additional insights on how the here presented cell types and insights are related to "classical neurotransmitters", other neuropeptides, and how the neuronal subtypes might interfere with other cells/cell types like Schwann cells, microglia, satellite glia.

We agree with the reviewer’s suggestion. As it opens a new field, and we will extend our current studies in this direction. To directly address this point, we have revised the discussion to more clearly state that NPY+ and NPY- neurons all have the machinery to make catecholamines (line 335). Thus, as indicated in the graphical abstract, cardiac neurons in stellate ganglia synthesize and release norepinephrine, however, a subset co-release NPY while others do not. Regarding interactions with other subtypes, this is an intriguing question. Drawing from CNS literature, astrocytes have shown circuit-specificity i.e. astrocytes associated with different brain circuits appear to exhibit some functional differences. This may be true in stellate ganglia as well, in terms of satellite glial cells and should be investigated.

References

Ajijola, O.A., Chatterjee, N.A., Gonzales, M.J., Gornbein, J., Liu, K., Li, D., Paterson, D.J., Shivkumar, K., Singh, J.P., and Herring, N. (2020). Coronary Sinus Neuropeptide y Levels and Adverse Outcomes in Patients with Stable Chronic Heart Failure. JAMA Cardiol. 5, 318–325. https://doi.org/10.1001/jamacardio.2019.4717.

Kalla, M., Hao, G., Tapoulal, N., Tomek, J., Liu, K., Woodward, L., Dall’Armellina, E., Banning, A.P., Choudhury, R.P., Neubauer, S., et al. (2020). The cardiac sympathetic co-transmitter neuropeptide Y is pro-arrhythmic following ST-elevation myocardial infarction despite β-blockade. Eur. Heart J. 41. https://doi.org/10.1093/eurheartj/ehz852.

Li, Y., and Dahlström, A. (2008). Peripheral projections of NESP55 containing neurons in the rat sympathetic ganglia. Auton. Neurosci. Basic Clin. 141. https://doi.org/10.1016/j.autneu.2008.03.008.

Rajendran, P.S., Challis, R.C., Fowlkes, C.C., Hanna, P., Tompkins, J.D., Jordan, M.C., Hiyari, S., Gabris-Weber, B.A., Greenbaum, A., Chan, K.Y., et al. (2019). Identification of peripheral neural circuits that regulate heart rate using optogenetic and viral vector strategies. Nat. Commun. 10. https://doi.org/10.1038/s41467-019-09770-1.

Tan, C.M.J., Green, P., Tapoulal, N., Lewandowski, A.J., Leeson, P., and Herring, N. (2018). The role of neuropeptide Y in cardiovascular health and disease. Front. Physiol. 9, 1–13. https://doi.org/10.3389/fphys.2018.01281.

Taniguchi, T., Morimoto, M., Taniguchi, Y., Takasaka, M., and Totoki, T. (1994). Cutaneous distribution of sympathetic postganglionic fibers from stellate ganglion: A retrograde axonal tracing study using wheat germ agglutinin conjugated with horseradish peroxidase. J. Anesth. 8. https://doi.org/10.1007/BF02514624.

Zhu, P., Sun, W., Zhang, C., Song, Z., and Lin, S. (2016). The role of neuropeptide Y in the pathophysiology of atherosclerotic cardiovascular disease. Int. J. Cardiol. 220, 235–241. https://doi.org/10.1016/j.ijcard.2016.06.138.

**Author response table 1. sa2table1:** Supplementary Table 1.

Sample	Batch	Sequencing Depth (UMIs/Cell)	Sequencing Saturation
Mouse-2–1_WTLM	2	5,558	60.9%
Mouse-2–2_WTLM	2	6,241	68.3%
Mouse-2–3_WTLM	2	3,818	83.1%
Mouse-2–4_WTLM	2	5,627	59.8%
Mouse-2–6_WTLM	2	4,953	60.4%
Mouse-1–3_WTLM	1	5,599	88.5%
Mouse-1–6_WTLM	1	4,441	87.8%
Mouse-1–7_WTLM	1	5,157	81.2%